# Intense training prevents the amnestic effect of inactivation of dorsomedial striatum and induces high resistance to extinction

**Martha Martínez-Degollado, Andrea C. Medina, Paola C. Bello-Medina, Gina L. Quirarte[ID], Roberto A. Prado-Alcalá[ID]***

Departamento de Neurobiología Conductual y Cognitiva, Instituto de Neurobiología, Universidad Nacional Autónoma de México, Querétaro, México

* prado@unam.mx

## Abstract

A large body of evidence has shown that treatments that interfere with memory consolidation become ineffective when animals are subjected to an intense learning experience; this effect has been observed after systemic and local administration of amnestic drugs into several brain areas, including the striatum. However, the effects of amnestic treatments on the process of extinction after intense training have not been studied. Previous research demonstrated increased spinogenesis in the dorsomedial striatum, but not in the dorsolateral striatum after intense training, indicating that the dorsomedial striatum is involved in the protective effect of intense training. To investigate this issue, male Wistar rats, previously trained with low, moderate, or high levels of foot shock, were used to study the effect of tetrodotoxin inactivation of dorsomedial striatum on memory consolidation and subsequent extinction of inhibitory avoidance. Performance of the task was evaluated during seven extinction sessions. Tetrodotoxin produced a marked deficit of memory consolidation of inhibitory avoidance trained with low and moderate intensities of foot shock, but normal consolidation occurred when a relatively high foot shock was used. The protective effect of intense training was long-lasting, as evidenced by the high resistance to extinction exhibited throughout the extinction sessions. We discuss the possibility that increased dendritic spinogenesis in dorsomedial striatum may underly this protective effect, and how this mechanism may be related to the resilient memory typical of post-traumatic stress disorder (PTSD).

## Introduction

Amnesia is produced by treatments that induce temporary disruption of normal electrical activity of brain structures involved in memory consolidation, such as tetrodotoxin (TTX) [1–8]. TTX blocks voltage-gated sodium channels, thus suppressing nerve impulses. There are cases, however, where memory is spared when this type of amnestic treatment is administered. For example, disruption of the electrical activity of the whole dorsal striatum [9,10], amygdala [10], hippocampus [4,11], and prelimbic area (PL) of the medial prefrontal cortex [12] impedes

**Data Availability Statement:** All relevant data are within the manuscript and its Supporting Information file.

**Funding:** RAP-A (IN205222) Dirección General de Asuntos del Personal Académico (DGAPA), Universidad Nacional Autónoma de México, https://dgapa.unam.mx/ RAP-A (CONACYT, 27570) Consejo Nacional de Ciencia y Tecnología, https://conahcyt.mx/ MM-D (Fellowship 298047) Consejo Nacional de Ciencia y Tecnología, https://conahcyt.mx/ The funders did not play any role in the study design, data collection and analysis, decision to publish, or preparation of the manuscript.

**Competing interests:** The authors have declared that no competing interests exist.

memory consolidation of low and moderate inhibitory avoidance (IA) training; nevertheless, increasing the intensity of the foot shock used for training protects memory consolidation from such disruption.

There is convincing evidence pointing to a functional distinction between dorsomedial (DMS) and dorsolateral (DLS) striatum, where DMS is predominantly involved in spatial and contextual memory while DLS deals mainly with stimulus-response and habit formation [13–21]. Because the one-trial step-through IA task involves the Pavlovian association of a context with a foot shock, it has been considered a spatial/contextual task; this task also entails the procedural components of receiving a foot shock upon crossing to the shock compartment, as well as an escape response (in our IA paradigm). Thus, the activity of both DMS and DLS seems necessary for memory consolidation of IA.

Some recent results, however, indicate that the DMS, but not the DLS, is importantly engaged in the protective effect of intense training against the effects of amnestic treatments. One piece of evidence was provided by the finding, mentioned above, that inactivation of PL produces amnesia after low and moderate, but not intense training [12]. Interestingly, PL has direct efferent connections with DMS and not with DLS [22–28].

Another piece of evidence was provided by Bello-Medina et al. [29], who reported that during memory consolidation of moderate IA training, there was a significant increase in the population of dendritic mushroom spines both in DLS and DMS; however, intense IA training further increased the density of these spines only in DMS. Recently, the same results were obtained when mushroom spinogenesis was quantified in DLS and DMS after retrieval of moderate and intense IA training [30]. It is worth mentioning that motor learning induces labeling of recently potentiated spines in the primary motor cortex of rats while shrinkage of these spines hinders the learned response, thus giving strong support to the hypothesis that spinogenesis represents a structural correlate of memory [31].

There is ample evidence that DLS is critically involved in extinction memory of stimulus-response and habit formation (e.g., [32–37]). There are, by contrast, only a few studies dealing with the involvement of DMS in extinction [37,38], and, to the best of our knowledge, the effects of interference with neural activity on extinction of intense learning have not been explored.

In the present work we studied the effects of TTX on memory consolidation and subsequent extinction of low, moderate, and intense IA training. Given the morphological evidence indicating that DMS is involved in memory consolidation of intense IA training, we predicted that functional inactivation circumscribed to DMS would hinder memory consolidation and resistance to extinction of low and moderate, but not intense IA training.

## Materials and methods

This study was carried out in strict accordance with the recommendations in the Guide for the Care and Use of Laboratory Animals of the National Institutes of Health [39], and all experimental procedures were approved by the Animal Ethics Committee of Instituto de Neurobiología, Universidad Nacional Autónoma de México (Protocol 098.A). Surgery was performed under sodium pentobarbital anesthesia, and all efforts were made to minimize animal´s suffering throughout the experiments. Animal management was supervised by a licensed veterinarian.

### Animals

We studied naïve adult male Wistar rats (250–350 g; 12–14 weeks old on arrival to the laboratory vivarium), obtained from the breeding colony of the Instituto de Neurobiología,

Universidad Nacional Autónoma de México; they were maintained in a room with a 12/12 h light-dark cycle (lights on at 7:00 h) and housed individually in acrylic cages with food and tap water *ad libitum*. The temperature of the room where the animals were housed was 23 ± 1˚C.

Before the experiments started, subjects received 5-min daily handling sessions for three consecutive days. All behavioral observations were carried out between 9:00 h and 14:00 h, the rats were randomly assigned to each group, and the experimenters were blind to treatments.

## Surgical procedures

The rats were anesthetized with sodium pentobarbital (50 mg/kg, i.p.), treated with atropine sulfate (0.4 mg/kg, i.p.), and positioned in a stereotaxic instrument (Stoelting Co., IL.). Stainless steel guide cannulae (23-gauge, 11-mm long) were implanted bilaterally into the DMS (anteroposterior + 0.4 mm relative to bregma; mediolateral ± 2.2 mm from midline; dorsoventral– 4.0 mm from skull surface) based on the coordinates from Paxinos and Watson [40]. The cannulae were affixed to the skull using one screw and dental cement. Stylets (11-mm long) were inserted into each cannula to maintain patency and were removed during the manipulation sessions and before the administration of treatments. After surgery, the animals received 1.0 ml of 0.9% saline solution (i.p.) to maintain hydration and facilitate the clearance of drugs and were kept in an incubator until fully recovered from anesthesia. The rats were allowed to recuperate seven days before initiation of training.

## Apparatus and behavioral procedures

A detailed procedure for training and testing the one-trial step-through IA task has been published elsewhere [29,41]. Briefly, the apparatus used for conditioning is an alley with two distinct compartments separated by a guillotine door. The safe compartment is illuminated, and the non-illuminated shock compartment has front and back walls and floor made of electrifiable stainless-steel plates. During training each rat was put inside the lit compartment, 10 s later the door between compartments was opened, and the latency to cross to the dark compartment was measured (training latency). When the animals crossed to this compartment the door was closed and a foot shock was delivered (foot shock intensities will be specified below). Five seconds after shock onset the door was reopened, allowing the animal to escape to the lit compartment (escape latency). Depending on the experimental design, some groups received one, two, or seven extinction sessions, starting 24 h after training; during these sessions memory of the task was measured as retention latencies. In these sessions, the same procedure as in training was followed except that the foot shock was not delivered. If a rat did not cross to the shock compartment within 600 s, the session ended, and a retention score of 600 was assigned (retention latency).

## Pharmacological treatments

All treatments were delivered in the form of simultaneous bilateral 0.5 μL infusions into the DMS through 30-gauge injection needles connected to a 10-μL Hamilton microsyringe by polyethylene tubing. The injection needles were inserted into the guide cannulae and protruded 1.0 mm beyond the tip of the cannulae. The infusion rate was 0.25 μL/min and was controlled by an automated microinfusion pump (WPI, model 220i). At the end of the infusion, the injection needles remained inside the guide cannulae for 60 s to minimize backflow. The injection procedure was carried out in a room different from that in which training and testing took place.

The dose of TTX reported to interfere with memory consolidation of IA, when infused into the striatum, is 10 ng dissolved in 1.0 μL of the vehicle solution, which affected a significant portion of the dorsal-ventral region of the intermediate striatum [5]. Because we wanted to

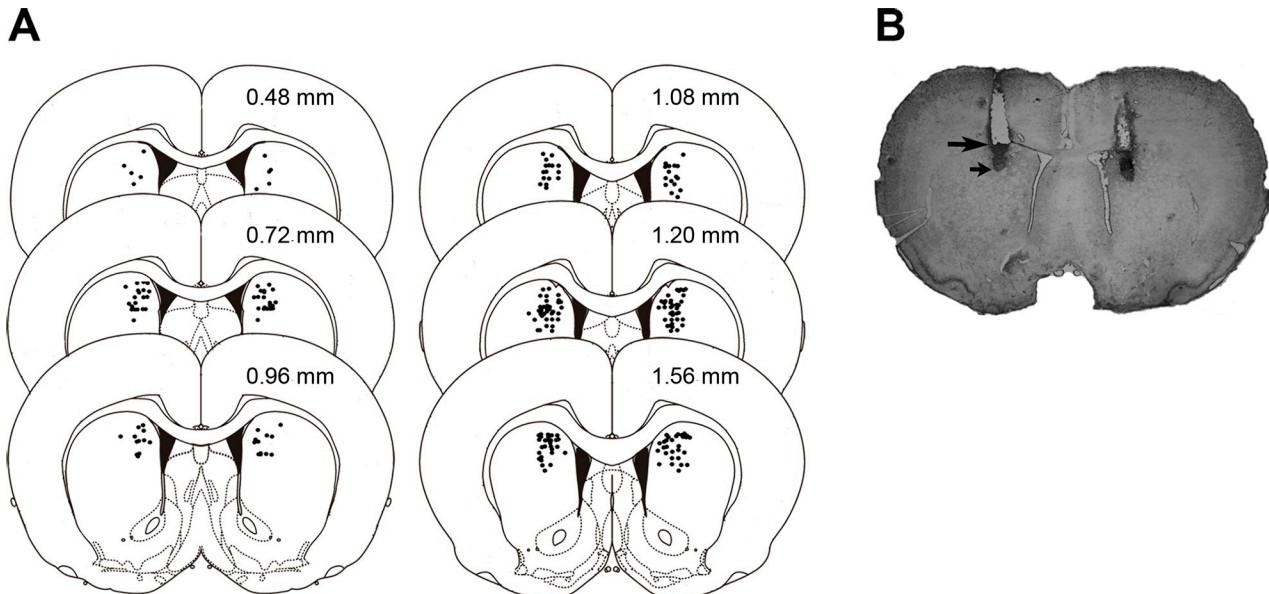

**Fig 1. Histological results.** (A) Diagrams illustrating the location of infusion needle tips in dorsomedial striatum of the rats included in the statistical analyses. The number of needle tips is less than the actual number because of overlapping sites. (B) The photomicrograph shows a representative coronal slice with the tracks left by the cannulae (large arrow) and injector needle tips (small arrow). Numbers in the diagrams represent distance from bregma. Redrawn from Paxinos and Watson [40].

disrupt the electrical activity of a restricted region of this structure, the DMS, we administered a smaller dose of TTX to reduce the possibility of spread to adjacent regions. Thus, 2.5 ng of TTX (Sigma–Aldrich, C11H17N3O8, T8024), dissolved in 0.5 µL of saline solution (0.90% w/v of NaCl) was used. Control rats were microinjected with the same volume of the vehicle solution (VEH). Because we wanted to train the animals while the activity of DMS was blocked, the microinjection of TTX was delivered 30 min before training; this duration was also used for the infusion of VEH. We had previously shown that this interval between TTX infusion into the hippocampus and training of IA produces a marked amnestic state [4,11]. The experimenters were blind to treatment allocation.

## Histology

The rats were anesthetized with sodium pentobarbital (125 mg/kg) and were perfused intracardially with 0.9% saline solution followed by a 4% formaldehyde solution. The brains were removed and immersed in the 4% formaldehyde solution for at least five days. Sections were cut (50 µm thick) on a cryostat and stained with cresyl-violet. The sections were examined under a light microscope, and the location of the injection needle tips was determined. Fig 1 shows a representation of injector tip locations within the DMS. The data from rats with incorrect placement of the cannula tips were not included in the statistical analyses. The number of animals used in each group is given in the Results section.

## Statistics

Independent Kruskal–Wallis ANOVAs were computed for training, escape, and retention latencies. When appropriate, the Mann–Whitney U test was used to make comparisons between any two groups. In the experiments on extinction the Friedman test was used to analyze retention latencies across the seven days of testing for each group, followed by the

Wilcoxon signed rank test to make comparisons between any two days along the extinction sessions in each group.

## Results

### Histology

As depicted in Fig 1, all the infusions of TTX and VEH in the animals that were included in the statistical analyses were made into the dorsomedial aspect of the striatum.

### Infusion of TTX into dorsomedial striatum disrupts memory consolidation

To find out if infusion of the relatively small dose of TTX (2.5 ng) into the DMS interferes with memory consolidation of IA trained with 1.0 mA, as evaluated in the first extinction session, one group of rats received the drug and another group received VEH; 24 h later retention of the task was measured. There were no significant differences between the groups regarding training and escape latencies (p = 0.07 and p = 0.39, respectively), but a significant deficiency in retention was produced by TTX (p < 0.0001) (Fig 2).

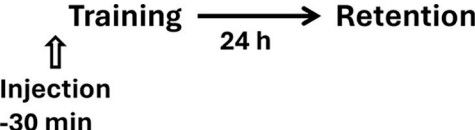

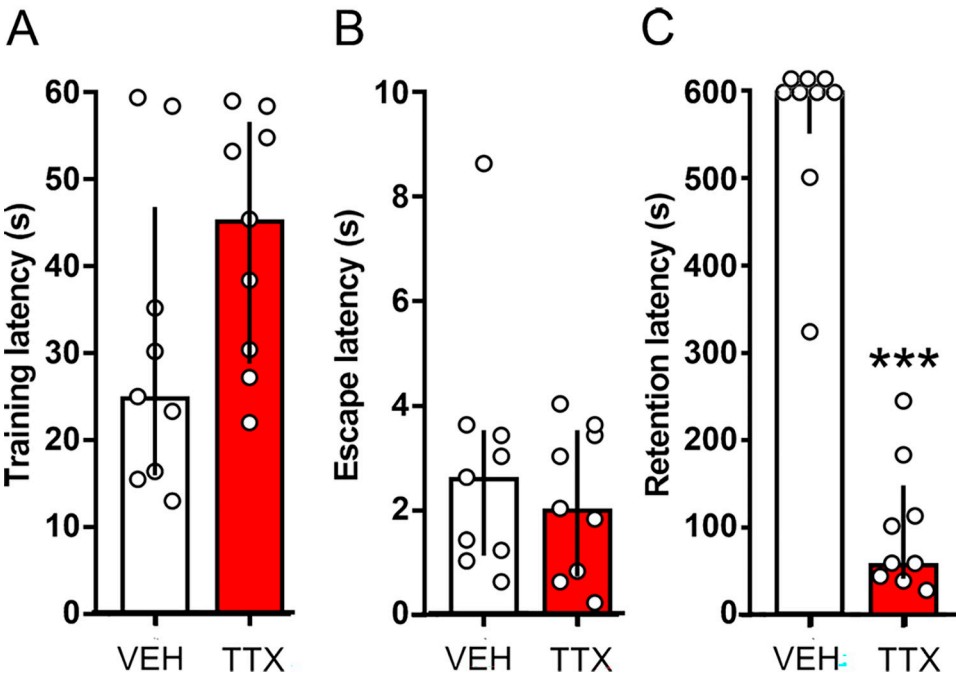

**Fig 2. Effect of pre-training infusions of TTX into the dorsomedial striatum on memory consolidation of one-trial step-through inhibitory avoidance trained with 1.0 mA.** Timeline of experimental procedure is shown on top of figure. Data represent median step-through latencies, in seconds, with interquartile ranges. There were no significant differences between the TTX and VEH groups in training (A) and escape (B) scores. A significant retention deficit was found after administration of TTX (C). In this and in the following figures, the small circles represent individual data. *** p < 0.0001 vs. VEH; n = 9 rats per group.

## TTX did not induce state-dependency

Because TTX was administered 30 min before training, it was possible that the retention deficit could have been caused by state dependency (because the animals were trained under the effect of TTX and tested in a drug-free condition), and not by interference with memory consolidation. To shed light on this issue one group received TTX and a second group received VEH twice: before training and before the 24-h retention test. Training was carried out using 1.0 mA. There were no significant differences in training and escape latencies between the groups ($p = 0.23$ and $p = 0.27$, respectively); in contrast, the VEH group had an optimal retention score whereas the TTX group displayed poor retention of the task ($p < 0.01$ vs. VEH) (Fig 3).

## TTX did not interfere with learning

This experiment evaluated whether TTX induced a learning deficit rather than a memory consolidation deficit. One group of rats received TTX, and another group received VEH; learning was measured 30 min after training with 1.0 mA and retention was evaluated 24 h after

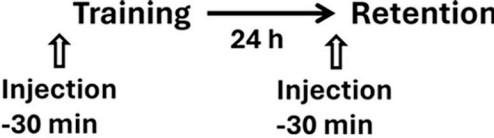

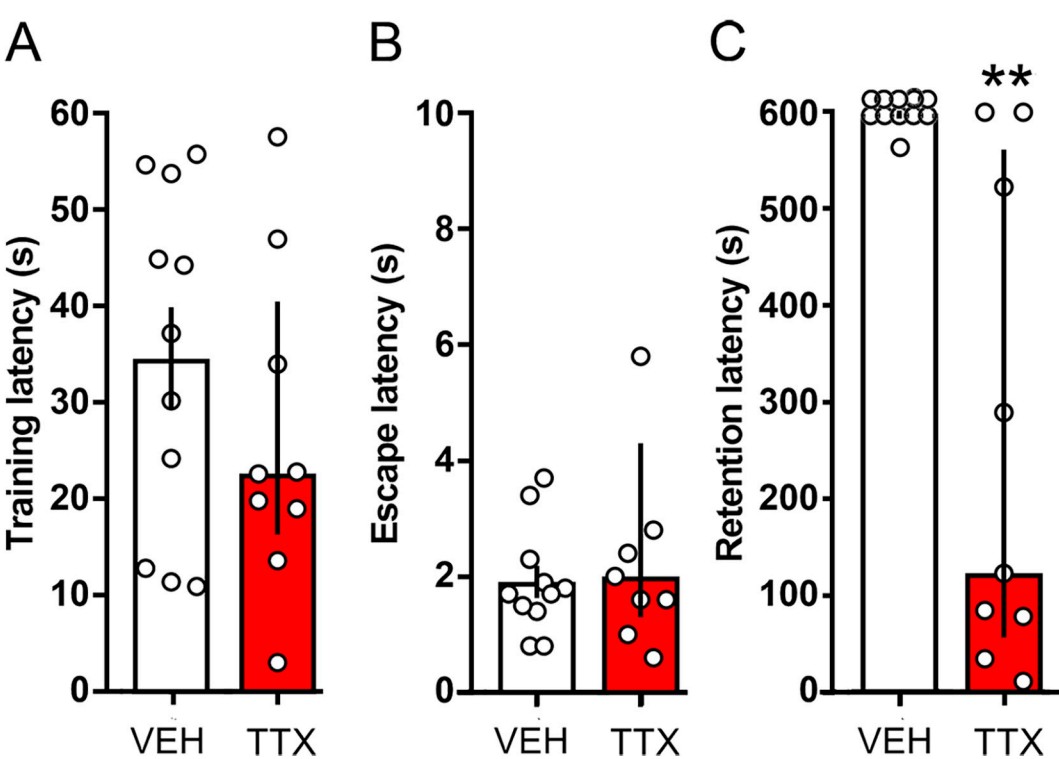

**Fig 3. Test for state-dependency.** Timeline of experimental procedure is shown on top of figure. Data represent median step-through latencies, in seconds, with interquartile ranges. TTX or VEH was infused into the dorsomedial striatum both before the training and retention sessions. Training was carried out using 1.0 mA. There were no significant differences between the TTX and VEH groups in training (A) and escape (B) scores, but a significant retention deficit was found after administration of TTX (C). ** $p < 0.01$ vs. VEH; number of rats per group: VEH = 11, TTX = 9.

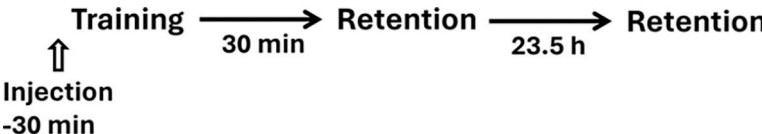

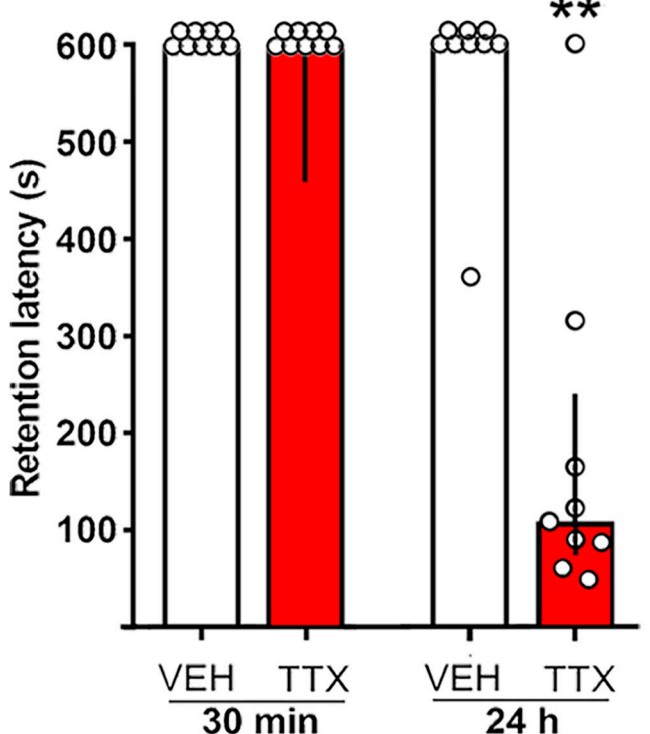

**Fig 4. Effect of pretraining TTX administration into the dorsomedial striatum on short-term and long-term retention latencies.** Timeline of experimental procedure is shown on top of figure. Data represent median step-through latencies, in seconds, with interquartile ranges. Training was carried out using 1.0 mA. There were no significant differences between the TTX and VEH groups in short-term memory (30 min) retention scores, but a significant long-term (24 h) retention deficit was produced by TTX. ** $p < 0.01$ vs. VEH; n = 9 per group. The number of groups in this experiment and in the remaining experiments was relatively high and for this reason in this figure and the following figures training and escape latencies were omitted to avoid overfilling. In all cases, training and escape latencies of the experimental groups did not differ significantly from VEH controls (Fig 4).

training in the same animals. In the 30-min test, there were no significant differences between these groups as both showed optimal retention scores ($p = 0.24$); in the 24-h test the VEH group maintained the previous score while the TTX group was amnestic ($p < 0.01$) (Fig 4).

## Intense training protects against the amnestic effect of TTX

Once we had found that infusion of TTX into DMS disrupted memory consolidation it was important to investigate whether intense training would impede its amnestic effect, as was reported after its administration into the dorsal hippocampus [4,11] and PL [12]. To study the effects of intense training, an operational definition of this variable was in order. Thus, we defined intense or strong IA training as that in which training with a relatively high foot shock intensity produces higher resistance to extinction than lower foot shock intensities. Resistance to extinction is a measure of the strength of learning because resistance to extinction is

stronger when the learning experience is also stronger; we had previously found that, in our hands, a foot shock of 3.0 mA fulfills this criterion [4,12,29,30,41–43,45].

We have studied the phenomenon of extinction using the same strain of rats, the same inhibitory avoidance conditioning chambers, and foot shock parameters. We have found that 3.0 mA induces stronger learning (and memory) than 2.0, 1.0, 0.5, and 0 mA; there are no significant differences between rats trained with 1.0 or 2.0 mA; in turn, the latter learn better than those trained with 0.5 mA; 0 mA does not produce IA learning. Based on these results we have termed low, moderate, and high or intense training as that mediated by 0.5 mA, 1.0 or 2.0 mA, and 3.0 mA, respectively [see references 4,12,29,30,41–43,45].

This experiment would establish whether the protective effect of intense training against the typical amnestic effect of TTX is limited to the dorsal hippocampus and PL, or whether it can be generalized to DMS. To this end, training of IA was carried out with 0.0, 0.5, 1.0, and 3.0 mA, and for each intensity there were two groups of rats: a TTX group and a VEH group. Starting 24 h after training, extinction of the task was evaluated on seven consecutive days, during which the foot shock was omitted. To better illustrate the protective effect of intense training, in this section we present the results obtained on the first extinction session, and in the following section the effects on the seven days of extinction will be described.

When comparing the retention scores of the TTX group against the scores of the VEH group at each intensity of training, the data revealed that there were no significant differences when 0.0 mA or 3.0 mA were administered (p = 0.08 and p = 0.24, respectively), and that amnesia was produced when the aversive stimulation was 0.5 or 1.0 mA (p < 0.0001 in both cases) (Fig 5).

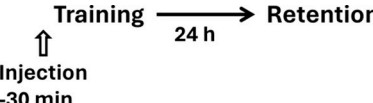

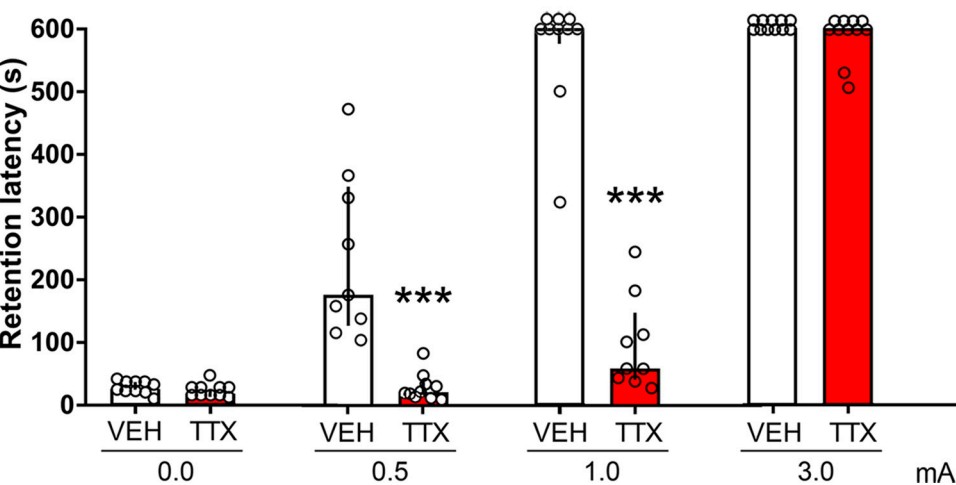

**Fig 5. Effects of pre-training infusions of TTX into the dorsomedial striatum on retention during the first extinction session of one-trial inhibitory avoidance trained with no foot shock (0.0 mA), or with low (0.5 mA), moderate (1.0 mA), and high (3.0 mA) foot shock intensities.** Timeline of experimental procedure is shown on top of figure. Data represent median step-through latencies, in seconds, with interquartile ranges. There were no significant differences between the VEH and TTX 0.0 mA groups. When compared with their respective control (VEH) group, a highly significant retention deficit was produced by TTX after training with 0.5 and 1.0 mA, but an optimal retention score was evident in the 3.0 mA group. *** p < 0.0001 vs. VEH; number of rats per group: VEH 0.0 mA = 10, TTX 0.0 mA = 10, VEH 0.5 mA = 9, TTX 0.5 mA = 10, VEH 1.0 mA = 10, TTX 1.0 mA = 9, VEH 3.0 mA = 11, TTX 3.0 mA = 11. The number of animals in each group is the same for Figs 6 and 7.

## Intense training produces high resistance to extinction after intra-DMS TTX administration

As described above, training of IA was carried out with 0.0, 0.5, 1.0, and 3.0 mA, and for each intensity there were two groups: a TTX group and a VEH group. Starting 24 h later, extinction of the task was evaluated on seven consecutive days, during which the foot shock was omitted.

With regard to the VEH groups, the Friedman test indicated that there was a significant effect of foot shock intensity across the extinction sessions on step-through latencies of the 0.0 mA group ($Q(6) = 16.68$, $p = 0.01$); the post-hoc Wilcoxon signed rank test revealed a small, significant reduction in the latency of the first "extinction" session relative to the latencies of the second ($p = 0.01$) and sixth ($p = 0.04$) "extinction" sessions. When analyzing the behavior of the 0.5 mA group, we also found significant differences among the extinction sessions ($Q(6) = 18.88$, $p < 0.01$); the latency of the first extinction session differed significantly from the latencies in each of the remaining six extinction sessions (p values ranging between $< 0.01$ and $< 0.05$). Similarly, there was a significant session effect in the 1.0 mA group ($Q(6) = 29.19$, $p < 0.0001$); the latency in the first extinction session differed significantly from the latencies of the rest of the sessions (p values ranging from $< 0.05$ to $< 0.005$). With respect to the 3.0 mA group, the Friedman test also yielded a significant session effect ($Q(6) = 42.78$, $p < 0.0001$); in this case, the latency score of the first extinction session only differed from the latency scores of the fourth, fifth, sixth, and seventh sessions (p values ranging from $< 0.05$ to $< 0.001$) (Fig 6).

The analyses of retention latencies of the TTX groups across the seven extinction sessions showed that there were no significant differences in the 0.0 and 0.5 mA groups ($Q(6) = 7.93$, $p = 0.24$ and $Q(6) = 6.35$, $p = 0.39$, respectively). There were significant changes in latency scores of the 1.0 mA group during the extinction sessions ($Q(6) = 16.43$, $p < 0.05$), where the

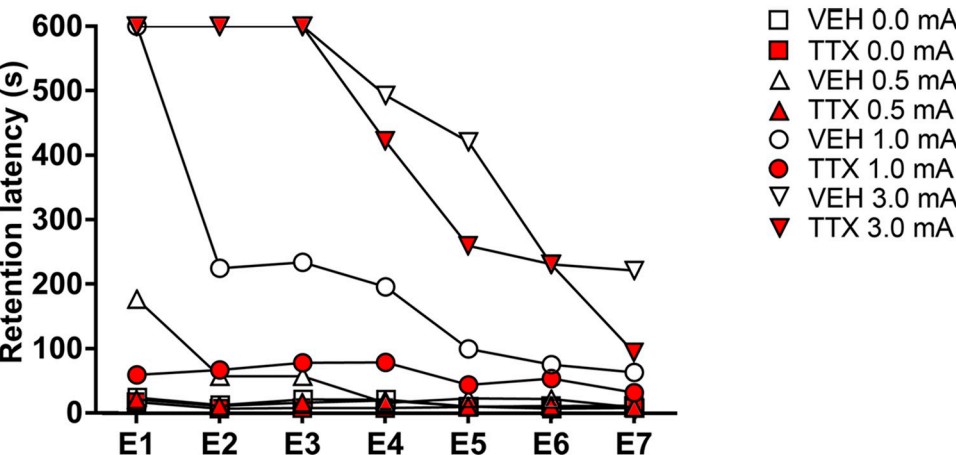

**Fig 6. Median retention latencies across the daily extinction sessions (E1–E7) shown by the groups trained in one-trial step-through inhibitory avoidance with 0.0, 0.5, 1.0, and 3.0 mA.** Timeline of experimental procedure is shown on top of figure. Note the high resistance to extinction displayed by the VEH and TTX groups that had been trained with 3.0 mA. See results of this section for details of statistical analyses.

first extinction session differed significantly from sessions 5, 6, and 7 ($p < 0.05$, 0.05, and 0.005, respectively). Lastly, the 3.0 mA group displayed significant differences during the course of the extinction sessions ($Q(6) = 48.5$, $p < 0.0001$). The post-hoc Wilcoxon test showed significant differences between the first extinction session and the fourth ($p < 0.05$), fifth ($p < 0.005$), sixth ($p < 0.001$), and seventh (0.0005) sessions (Fig 6).

To visualize better the effect of foot shock intensity upon resistance to extinction after the treatments, we grouped together the latency scores of the seven extinction sessions at each foot shock intensity for each of the TTX and VEH groups, thus yielding two sets of data per foot shock intensity, as seen in Fig 7. Comparisons between the VEH and TTX groups did not reveal significant differences in the 0.0 mA groups ($p = 0.07$), as they did not receive the aversive stimulation during "training". Significant differences became evident at 0.5 and 1.0 mA ($p < 0.01$ and $< 0.0001$, respectively). By contrast, there were no significant differences between the VEH and TTX groups that had been trained with 3.0 mA ($p = 0.38$). These results show that the highest foot shock intensity impeded the typical impairing impact of TTX on retention (Fig 7). Regarding resistance to extinction, the Kruskal-Wallis test showed a significant foot shock intensity effect in the VEH groups ($H(3) = 173.22$, $p < 0.0001$). The post-hoc Mann-Whitney test revealed that the 0.0 mA group had a significantly lower retention score than the 0.5, 1.0, and 3.0 mA groups ($p < 0.001$, $< 0.0001$, and $< 0.0001$, respectively); the 0.5 mA group also differed significantly from the 1.0 and 3.0 mA groups ($p < 0.0001$ for both comparisons); finally, the 1.0 mA group differed significantly from the 3.0 mA group ($p < 0.0001$). These results confirm that the highest intensity produced a stronger resistance to extinction than the lower intensities. Foot shock intensity also had an impact on the resistance to extinction displayed by the TTX groups ($H(3) = 167.43$, $p < 0.0001$). Pairwise comparisons showed that the 0.0 mA group had a significantly lower retention score than the 0.5, 1.0, and 3.0 mA groups ($p < 0.05$, $< 0.0001$, and $< 0.0001$, respectively); the 0.5 mA group differed

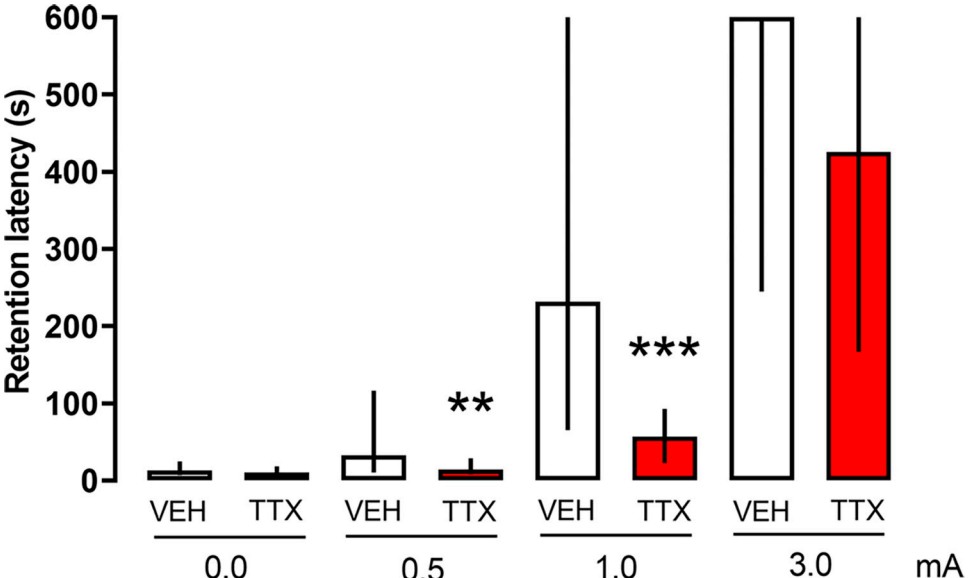

**Fig 7. Effect of pre-training infusions of TTX into the dorsomedial striatum on extinction of one-trial inhibitory avoidance trained with 0.0 mA, low (0.5 mA), moderate (1.0 mA), and high (3.0 mA) foot shock intensities.** The bars represent the median of latency scores (with interquartile ranges) of the seven extinction sessions for each group. When compared with their respective control (VEH) group, a significant retention deficit was produced by TTX after training with 0.5 and 1.0 mA, but no significant differences in retention scores appeared between the groups that had been trained with 0.0 or 3.0 mA. ** $p < 0.01$, *** $p < 0.0001$ vs. VEH.

significantly only from the 3.0 mA group (p < 0.0001); finally, the 1.0 mA group differed significantly from the 3.0 mA group (p < 0.0001). These data show that, just as in the case of the treatment with VEH, the higher foot shock intensity produced a higher resistance to extinction than the lower foot shock intensities (Fig 7).

## Discussion

There were two major findings in this study. First, interference with neuronal activity circumscribed to DMS, induced before training, produced a marked deficit of memory consolidation of IA trained with foot shocks of low and moderate intensities. We had previously described similar effects after inactivation of the whole dorsal striatum [10,43,44]. There are, however, several potential problems derived from the administration of treatments before training of IA that might lead to misinterpretation of the results obtained after low and moderate training. For example, the retention deficit could have occurred not because memory consolidation was hindered but, rather, because of a state-dependent effect that might have taken control over the learned behavior, because the animals had been trained under the influence of TTX and the test of memory was conducted in a drug-free state. The results of the test for state-dependency showed that such was not the case. Another possibility was that the drug did not allow for learning of the task and for this reason there was a deficient performance when long-term memory was evaluated. This possibility can also be discarded because when retention was measured 30 min after training, the TTX group had an optimal retention score; this group, however, was amnestic on the 24-h retention test. This finding also indicates that the activity of DMS is not necessary for the development of short-term memory. Finally, because in all the experiments of this study no significant differences were found in training and escape latencies between the TTX and VEH groups, it can be inferred that the relatively small dose of TTX used did not produce sensory, perceptual, or motor incapacities that might have affected the ability required to learn or perform the IA task.

The second major result of this study was that the TTX-induced interference with memory described above was prevented when the strength of training was increased. This protective effect was evident not only on the retention test run 24 h after training but also during the evaluation of extinction behavior. The TTX groups that had been trained with 0.5 and 1.0 mA displayed significantly lower step-through latencies than their respective VEH controls throughout the seven extinction sessions. This result indicates that the disruption of neural activity of DMS produced by TTX impeded memory consolidation when training was carried out with the low foot shock intensity (0.5 mA), and only allowed for the development of a fragile memory trace when the 1.0 mA foot shock was used. When the animals were trained with the higher aversive stimulation (3.0 mA) there were no significant differences in latency scores between the VEH and TTX groups, both of which showed optimal latency scores and stronger resistance to extinction than the rest of the groups.

In previous work, we interpreted the protective effect of intense training on memory consolidation to mean that intense training induces transfer of information to a wider neuronal network and the affected structures are no longer critical sites for the encoding mediating overtraining [4,9,29]. In agreement with this interpretation, recent findings from our laboratory have provided compelling evidence that intense training induces the recruitment of a higher number of neurons in the amygdala [42] and ventral striatum [45] which may contribute to the formation of a stronger memory trace. The present results, and those showing increased spinogenesis and recruitment of more neurons, are important steps in understanding how strong memory traces resulting from intense learning experiences are formed and may shed light into the genesis of pathologic memory disturbances, such as PTSD.

The protective effect of intense training against memory deficits produced by typical amnestic treatments is a robust phenomenon (for references see [43,44]); however, its underlying neurobiological foundations remain unclear. Recent evidence indicates that intense IA training induces enhanced proliferation of mushroom dendritic spines in DMS during the processes of memory consolidation [29] and memory retrieval [30]. Along this line, TTX inactivation of PL, which has monosynaptic connections with DMS [22–28], interferes with consolidation of moderate, but not with intense IA training [12]. Importantly, there are not only anatomical connections between PL and DMS, but functional connections as well. For example, Guo et al. [46] found a significant functional correlation between these two regions during a skilled task, and Friedman et al. [22] and Shipman et al. [47] described, respectively, that PL and DMS collaborate in the control of decision-making and in the performance of instrumental conditioning. Thus, it is conceivable that under physiological conditions intense training facilitates neuronal activity of PL, more than moderate or low levels of training, which through glutamate release from its efferent projections to DMS enhances mushroom spinogenesis, thus giving rise to a stronger memory trace. This possibility is worth investigating.

But how, then, would the DMS be involved in memory consolidation of intense IA training when its electrical activity had been disrupted by the TTX? One possibility is that even under this adverse condition, neurotransmitter release from PL terminals could activate membrane receptors located on the post-synaptic dendritic spines promoting augmented spinogenesis. We think that this is an unlikely possibility because the sodium channels of the efferent axons reaching DMS would be blocked by the TTX and, consequently, neurotransmitter release would be interrupted. Preliminary data from our laboratory indicates that as a result of intense training new spines can be produced in DMS despite its inactivation by TTX. Therefore, there must be a different mechanism leading to dendritic spine production in the inactivated DMS.

A reasonable possibility rests on sound evidence showing that aversive training in rodents induces the release of the glucocorticoid hormone corticosterone (CORT), which is dependent upon the intensity of the aversive stimulation during learning [45,48,49]. Moreover, infusion of CORT or CORT-receptor agonists into the striatum, amygdala, and hippocampus facilitates memory consolidation of IA [50–54].

It has also been shown that CORT administration increases the density of hippocampal dendritic spines *in vitro* in a dose-dependent manner. This effect was resistant to protein synthesis inhibitors and blocked by inhibitors of kinases such as PKA, PKC, and MAPK, suggesting non-genomic spinogenesis [55–59]. CORT-induced spinogenesis has also been described in the cortex *in vivo* [60], and dendritic spinogenesis induced by motor skill learning was promoted at the peak of circadian CORT release through a non-genomic mechanism [61]. To the best of our knowledge, the effects of CORT described above are independent of action potential activity.

This body of evidence leads us to propose a mechanism by which intense training protects memory against the amnestic effect of blockade of DMS electrical activity. We suggest that the increased release of CORT after intense training [45,48,49] induces augmented spinogenesis. Thus, binding of CORT to its cytoplasmatic receptor (and its subsequent translocation to the nucleus), and/or to a membrane-bound receptor (and the subsequent non-transcriptional activation of different kinase signaling pathways) leads to the genesis of more dendritic spines, where information of the learning experience might be stored.

One of the main symptoms of posttraumatic stress disorder is the long-lasting memory for traumatic life events, evidenced by the high resistance to extinction of such memory. In many cases, a single experience of the traumatic event is sufficient to produce this illness [62,63]. Since these features parallel those found after intense IA training, a possible neurobiological basis of the relentless memory PTSD patients may also be related to enhanced spinogenesis. This possibility merits being investigated.

## Conclusions

We had predicted that functional inactivation circumscribed to DMS would impede memory consolidation and impair subsequent extinction of low and moderate, but not intense IA training. In agreement with this prediction, we found that intense inhibitory avoidance training offsets the memory consolidation deficit produced by inactivation of DMS after low or moderate training and strengthens resistance to extinction. We propose that the protective effect of intense training is mediated by activation of plasmatic and/or membrane-bound corticosterone receptors. We are currently exploring these possibilities.

## Supporting information

**S1 Raw data.**
(PDF)

## Acknowledgments

The authors thank Norma Serafín, Bernardino Osorio, Nuri Aranda, Alejandra Castilla, Martín García, Maria E. Rosas, Omar González, and Ramón Martínez for their excellent technical and administrative assistance, and Dr. Michael C. Jeziorski for helpful comments on the manuscript. Martha Martínez-Degollado is a doctoral student of Programa de Doctorado en Ciencias Biomédicas, Universidad Nacional Autónoma de México.

## Author Contributions

**Conceptualization:** Martha Martínez-Degollado, Roberto A. Prado-Alcalá.

**Data curation:** Martha Martínez-Degollado, Andrea C. Medina, Paola C. Bello-Medina.

**Formal analysis:** Martha Martínez-Degollado, Andrea C. Medina, Paola C. Bello-Medina.

**Investigation:** Martha Martínez-Degollado, Roberto A. Prado-Alcalá.

**Methodology:** Martha Martínez-Degollado, Roberto A. Prado-Alcalá.

**Project administration:** Martha Martínez-Degollado, Andrea C. Medina, Paola C. Bello-Medina.

**Resources:** Gina L. Quirarte, Roberto A. Prado-Alcalá.

**Supervision:** Roberto A. Prado-Alcalá.

**Validation:** Andrea C. Medina, Paola C. Bello-Medina.

**Visualization:** Martha Martínez-Degollado, Andrea C. Medina, Paola C. Bello-Medina.

**Writing – original draft:** Roberto A. Prado-Alcalá.

**Writing – review & editing:** Martha Martínez-Degollado, Andrea C. Medina, Paola C. Bello-Medina, Gina L. Quirarte, Roberto A. Prado-Alcalá.

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
