## [Decision Letter · Decision Letter 0]

4 Apr 2024

PONE-D-24-06106Intense training prevents the amnestic effect of inactivation of dorsomedial striatum and induces high resistance to extinctionPLOS ONE

Dear Dr. Prado-Alcalá,

Thank you for submitting your manuscript to PLOS ONE. After careful consideration, we feel that it has merit but does not fully meet PLOS ONE’s publication criteria as it currently stands. Therefore, we invite you to submit a revised version of the manuscript that addresses the points raised during the review process.

We look forward to receiving your revised manuscript.

Kind regards,

Yukiori Goto, Ph.D.

Academic Editor

PLOS ONE

Journal Requirements:

2. To comply with PLOS ONE submissions requirements, in your Methods section, please provide additional information regarding the experiments involving animals and ensure you have included details on (1) methods of sacrifice, and (2) efforts to alleviate suffering.

Reviewers' comments:

Reviewer's Responses to Questions

**Comments to the Author**

1. Is the manuscript technically sound, and do the data support the conclusions?

Reviewer #1: Yes

Reviewer #2: Yes

2. Has the statistical analysis been performed appropriately and rigorously? 

Reviewer #1: Yes

Reviewer #2: Yes

3. Have the authors made all data underlying the findings in their manuscript fully available?

Reviewer #1: No

Reviewer #2: Yes

4. Is the manuscript presented in an intelligible fashion and written in standard English?

Reviewer #1: Yes

Reviewer #2: Yes

5. Review Comments to the Author

Reviewer #1: Roberto et al. investigated the effect of intense training on memory and extinction. They reported that intense training induces resistance to the extinction.

I believed this manuscript is good but there are several concerned that should be attention by authors.

The main point is that this finding should be support by histological parameters. I think neurological index such as Neru-D.

Why did the researchers not use other learning tests such as the Morris Water Maze?

How did the researchers divide the intensity into low, medium and high? Is there a standard?

Point 1: PTSD abbreviation should be explaining previously.

Point 2: It is surprising that a 9-week-old animal has a 350 g weight.

Point 3: number of rats in each group and study design should be mention in materials and methods.

Point 4: The quality of figure 1 is not clear.

Reviewer #2: This study examined the effects of TTX inactivation of dorsomedial striatum in memory consolidation of foot shocks.

TTX impaired consolidation with low intensities of foot shocks , but not with the high intensity, which are highly interesting.

The major shortcoming of the study is obviously a lack of explanation for the underlying mechanisms, although the authors have proposed a highly speculative idea that spinogensis may be involved in the observations.

Alongside this issue, I have only a few minor concerns.

The word "intense training" has been used repeatedly in the title and over the manuscript, which misleads the study, as the study examined different intensities of foot shock, but not intensive training itself. Thus, it is better to replace the word with a more appropriately describing the experimental condition.

It would be helpful to include a diagram describing the experimental design and timeline of drug treatments and behavioral training/testing.

For instance, data in Figure 5 and 7 have reported the data alike, but the experiment was conducted differently for each of them are obscure.

Please indicate whether the error bars in the figures are SD or SEM.

Line 47-48

The sentence "Hindrance of memory consolidation is produced by ... tetrodotoxin" can read as if taking TTX causes amnesia, which is actually not. This should state more accurately that infusion of TTX to inactivate a specific brain region could cause amnesia.

6. PLOS authors have the option to publish the peer review history of their article (what does this mean?). If published, this will include your full peer review and any attached files.

Reviewer #1: No

Reviewer #2: No

---

## [Author Response · Author response to Decision Letter 0]

17 May 2024

Reviewer's Responses to Questions

Comments to the Author

1. Is the manuscript technically sound, and do the data support the conclusions?

Reviewer #1: Yes

Reviewer #2: Yes

2. Has the statistical analysis been performed appropriately and rigorously?

Reviewer #1: Yes

Reviewer #2: Yes

3. Have the authors made all data underlying the findings in their manuscript fully available?

Reviewer #1: No

Reviewer #2: Yes

Reply:

All the data underlying our findings are fully available in the file we uploaded in our original submission (Supporting Information; File Name: S1 DATA.pdf).

4. Is the manuscript presented in an intelligible fashion and written in standard English?

Reviewer #1: Yes

Reviewer #2: Yes

5. Review Comments to the Author

Reviewer #1: Roberto et al. investigated the effect of intense training on memory and extinction. They reported that intense training induces resistance to the extinction. I believed this manuscript is good but there are several concerned that should be attention by authors.

The main point is that this finding should be support by histological parameters. I think neurological index such as Neru-D.

Reply.

We tried hard to find the suggested neurological index Neru-D with no success; PubMed, Scopus, and other sources yielded 0 positive results. If this reviewer is asking about the injection sites, Figure 1 shows the location of the injector tips, all of which were within the dorsomedial striatum.

Why did the researchers not use other learning tests such as the Morris Water Maze?

Reply

This study aimed to investigate the effects of the inactivation of the dorsomedial striatum on memory consolidation and subsequent extinction of learning acquired with different degrees of training. To this end, we decided to use a behavioral task, the inhibitory avoidance task, that has been used in my laboratory for many years. One of the advantages of this procedure is that it entails only one trial and one aversive stimulus, thus allowing to determine with precision the duration of the consolidation process; furthermore, we have defined the parameters of aversive stimulation that yield those different degrees of training and the characteristics of extinction learning. Hence, this task is quite appropriate to answer our experimental questions. 

In the case of the Morris Water Maze task, on the other hand, rats require several trials and sessions to learn, so it is difficult to determine when consolidation takes place; the other reason we did not use this task is that it would take us a significant amount of time to establish the correct parameters to produce distinct degrees of training and extinction curves for each training condition. 

How did the researchers divide the intensity into low, medium, and high? Is there a standard?

Reply

In lines 263-270 of the revised manuscript, we have now defined low, moderate, and intense training as follows: “We have studied the phenomenon of extinction using the same strain of rats, inhibitory avoidance conditioning chambers, and footshock parameters. We have found that 3.0 mA induces stronger learning (and memory) than 2.0, 1.0, 0.5, and 0 mA; rats trained with 1.0 or 2.0 mA learn better than those trained with 0.5 mA; 0 mA does not produce IA learning. We measured the strength of learning through resistance to extinction (resistance to extinction is stronger when the learning experience is also stronger). Based on these results we have termed low, moderate, and intense training as that mediated by 0.5 mA, 1.0 or 2.0 mA, and 3.0 mA, respectively (see references 12, 29, 30, 42, and 45).” In the revised manuscript we have also shown the comparison of resistance to extinction among the control groups; resistance to extinction was statistically higher in the 3.0 mA group, followed by the 1.0 mA group, and then by the 0.5 mA group.” Lines 345-353.

Point 1: PTSD abbreviation should be explaining previously.

Reply

We thank the reviewer´s comment. We have now defined PTSD as post-traumatic stress disorder (line 43).

Point 2: It is surprising that a 9-week-old animal has a 350 g weight.

Reply

We checked the age of the rats used in this study. They were, actually, 12-14 weeks old; we have made this correction in the new manuscript (line 100).

Point 3: number of rats in each group and study design should be mention in materials and methods.

Reply

The number of rats varies in each experiment, so, in the original manuscript the sample size of each group was specified in the Figure legend of each experiment. 

Point 4: The quality of figure 1 is not clear.

Reply

The PDF version of the original Figure 1, made by the Journal, looks like this:

However, the original figure is the following one, clearly showing the principal details. In case it is accepted, this is the figure that would be presented in our article.

Reviewer #2: This study examined the effects of TTX inactivation of dorsomedial striatum in memory consolidation of foot shocks. TTX impaired consolidation with low intensities of foot shocks, but not with the high intensity, which are highly interesting.

The major shortcoming of the study is obviously a lack of explanation for the underlying mechanisms, although the authors have proposed a highly speculative idea that spinogensis may be involved in the observations.

Reply

In the Discussion of the original manuscript, based on experimental results from other laboratories as well as our laboratory, we suggested several possible mechanisms that may underly the protective effect of high-intensity aversive stimulation training on memory consolidation; these suggested mechanisms are:

 A. Intense training induces the transfer of information to a wider neuronal network, and the affected structures are no longer critical sites for the encoding of overtraining [4,9,29]. In agreement with this interpretation, recent findings from our laboratory have provided compelling evidence that intense training induces the recruitment of a higher number of neurons in the amygdala [42] and ventral striatum [45] which may contribute to the formation of a stronger memory trace (lines 406-412).

B. There is sound evidence showing that aversive training in rodents induces the release of the glucocorticoid hormone corticosterone (CORT), which is dependent upon the intensity of the aversive stimulation during learning [45, 48, 49]. Moreover, infusion of CORT or CORT-receptor agonists into the striatum, amygdala, and hippocampus facilitates memory consolidation of IA [50-54] (lines 443-447). It has also been shown that CORT administration increases the density of hippocampal dendritic spines in vitro in a dose-dependent manner. 

C. Preliminary data from our laboratory indicates that as a result of intense training new spines can be produced in DMS despite its inactivation by TTX (lines 439-441).

Thus, based on these data, we have proposed that the protective effect of intense training is mediated by activation of plasmatic and/or membrane-bound corticosterone receptors. We are currently exploring these possibilities.

Alongside this issue, I have only a few minor concerns.

The word "intense training" has been used repeatedly in the title and over the manuscript, which misleads the study, as the study examined different intensities of foot shock, but not intensive training itself. Thus, it is better to replace the word with a more appropriately describing the experimental condition.

Reply

We agree that “intense training” might lead to some confusion. To avoid this possibility, in the original manuscript we had given an operational definition of intense training, that we have used over the years, as seen in lines 254-260 in the original manuscript “We defined intense or strong IA training as that in which training with a relatively high foot shock intensity produces higher resistance to extinction than lower foot shock intensities. Resistance to extinction is a measure of the strength of learning because resistance to extinction is stronger when the learning experience is also stronger; we had previously found that, in our hands, a foot shock of 3.0 mA fulfills this criterion [29, 41, 42]”; we have added more references in support of this definition (4, 12, 43, 45) (lines 261-262).

In the revised manuscript we have also defined, operationally, these three levels of training, and have compared resistance to extinction among the control groups; resistance to extinction was statistically higher in the 3.0 mA group, followed by the 1.0 mA group, and then by the 0.5 mA group (lines 345-353).

It would be helpful to include a diagram describing the experimental design and timeline of drug treatments and behavioral training/testing. For instance, data in Figure 5 and 7 have reported the data alike, but the experiment was conducted differently for each of them are obscure.

Reply

We have now included a diagram describing the experimental design and timeline of drug treatments and behavioral training/testing in all the figures showing behavioral results.

Please indicate whether the error bars in the figures are SD or SEM.

Reply 

In all the figures where error bars were represented (Figures 2, 3, 4, 5, and 7), it was stated that the data represent median step-through latencies, in seconds, with interquartile ranges. 

Line 47-48.

The sentence "Hindrance of memory consolidation is produced by ... tetrodotoxin" can read as if taking TTX causes amnesia, which is actually not. This should state more accurately that infusion of TTX to inactivate a specific brain region could cause amnesia.

Reply

We fully agree with this reviewer and have changed this sentence to “Amnesia is produced by treatments that induce temporary disruption of normal electrical activity of brain structures involved in memory consolidation, such as tetrodotoxin (TTX) [1-8]” (lines 47-49).

6. PLOS authors have the option to publish the peer review history of their article (what does this mean?). If published, this will include your full peer review and any attached files.

Do you want your identity to be public for this peer review? For information about this choice, including consent withdrawal, please see our Privacy Policy.

Reviewer #1: No

Reviewer #2: No

---

## [Editor Report · Decision Letter 1]

23 May 2024

Intense training prevents the amnestic effect of inactivation of dorsomedial striatum and induces high resistance to extinction

PONE-D-24-06106R1

Dear Dr. Prado-Alcalá,

We’re pleased to inform you that your manuscript has been judged scientifically suitable for publication and will be formally accepted for publication once it meets all outstanding technical requirements.

Kind regards,

Yukiori Goto, Ph.D.

Academic Editor

PLOS ONE

Additional Editor Comments (optional):

Thank you for a revision on the manuscript, which has adequatly addressed the issues ratised by the referees at the initinal submission.
---

## [Editor Report · Acceptance letter]

28 May 2024

PONE-D-24-06106R1 

PLOS ONE

Dear Dr. Prado-Alcalá, 

I'm pleased to inform you that your manuscript has been deemed suitable for publication in PLOS ONE. Congratulations! Your manuscript is now being handed over to our production team.

Kind regards, 

on behalf of

Dr. Yukiori Goto 

Academic Editor

PLOS ONE